# Effects of Dietary Protein and Fat Content on Intrahepatocellular and Intramyocellular Lipids during a 6-Day Hypercaloric, High Sucrose Diet: A Randomized Controlled Trial in Normal Weight Healthy Subjects

**DOI:** 10.3390/nu11010209

**Published:** 2019-01-21

**Authors:** Anna Surowska, Prasanthi Jegatheesan, Vanessa Campos, Anne-Sophie Marques, Léonie Egli, Jérémy Cros, Robin Rosset, Virgile Lecoultre, Roland Kreis, Chris Boesch, Bertrand Pouymayou, Philippe Schneiter, Luc Tappy

**Affiliations:** 1Department of Physiology, University of Lausanne, 1005 Lausanne, Switzerland; anna.surowska@unil.ch (A.S.); pira_jegatheesan@hotmail.com (P.J.); vanessacaroline.campos@rdls.nestle.com (V.C.); asophie.marques@gmail.com (A.-S.M.); Leonie.Egli@rdls.nestle.com (L.E.); Jeremy.cros@unil.ch (J.C.); rosset.robin@gmail.com (R.R.); virgile.lecoultre@hibroye.ch (V.L.); philippe.schneiter@unil.ch (P.S.); 2Department for Biomedical Research, University of Bern and Institute of Diagnostic Interventional and Pediatric Radiology, University Hospital, 3012 Bern, Switzerland; roland.kreis@insel.ch (R.K.); Chris.Boesch@insel.ch (C.B.); bertrand.pouymayou@insel.ch (B.P.)

**Keywords:** sucrose overfeeding, hepatic steatosis, intramyocellular lipids, intrahepatocellular lipids, dietary protein content, dietary fat content, energy expenditure, plasma triglyceride

## Abstract

Sucrose overfeeding increases intrahepatocellular (IHCL) and intramyocellular (IMCL) lipid concentrations in healthy subjects. We hypothesized that these effects would be modulated by diet protein/fat content. Twelve healthy men and women were studied on two occasions in a randomized, cross-over trial. On each occasion, they received a 3-day 12% protein weight maintenance diet (WM) followed by a 6-day hypercaloric high sucrose diet (150% energy requirements). On one occasion the hypercaloric diet contained 5% protein and 25% fat (low protein-high fat, LP-HF), on the other occasion it contained 20% protein and 10% fat (high protein-low fat, HP-LF). IHCL and IMCL concentrations (magnetic resonance spectroscopy) and energy expenditure (indirect calorimetry) were measured after WM, and again after HP-LF/LP-HF. IHCL increased from 25.0 ± 3.6 after WM to 147.1 ± 26.9 mmol/kg wet weight (ww) after LP-HF and from 30.3 ± 7.7 to 57.8 ± 14.8 after HP-LF (two-way ANOVA with interaction: *p* < 0.001 overfeeding x protein/fat content). IMCL increased from 7.1 ± 0.6 to 8.8 ± 0.7 mmol/kg ww after LP-HF and from 6.2 ± 0.6 to 6.9 ± 0.6 after HP-LF, (*p* < 0.002). These results indicate that liver and muscle fat deposition is enhanced when sucrose overfeeding is associated with a low protein, high fat diet compared to a high protein, low fat diet.

## 1. Introduction

Consumption of hypercaloric high-fructose or high-sucrose diets can lead to the deposition of fat in ectopic sites such as visceral adipose tissue, the liver (intrahepatocellular lipids, IHCL), skeletal muscle (intramyocellular lipids, IMCL), the heart, and the pancreas [1]. Such ectopic fat deposition has been associated with insulin resistance and increased risk of cardiovascular and hepatic disorders [2,3]. In addition, hypercaloric high-fructose diets have been shown to impair hepatic insulin sensitivity [4,5], to increase fasting and postprandial blood triglycerides [6,7] and uric acid [8] concentrations, and may therefore be associated with a particularly ominous constellation of cardiometabolic risk factors.

Most studies that have documented metabolic effects of fructose or sucrose overfeeding have involved either the addition of fructose or sucrose to a weight maintenance diet, or the substitution of fructose or sucrose for dietary starch. In real life conditions, however, the addition of sucrose to an *ad libitum* diet is expected to impact habitual food consumption and hence to alter both total energy intake and the dietary macronutrient composition. It has indeed been reported that the addition of fructose-sweetened beverages to the spontaneous diet of overweight subjects was associated with a partial suppression of dietary fat and protein intake from solid foods [9]. One may therefore hypothesize that the metabolic effects of overfeeding depend not only on the amount of excess sucrose, but also on how it impacts other dietary macronutrient intake. Dietary sucrose and fat content may have additive effects on IHCL [10]. Interactions between dietary sucrose and protein are also relevant, since dietary protein intake has been shown to modulate overfeeding-induced ectopic lipid storage: in rodents fed a high fructose diet, the increase in IHCL was lower when excess dietary fructose was associated with a high, compared to a low, protein intake [11,12]. Similar observations were reported for humans overfed with lipids and protein compared to lipids alone [13,14,15], and with fructose and essential amino-acids compared with fructose alone [16]. In addition, a high protein intake is associated with an increase in energy expenditure, and may thus reduce energy storage [17]. We therefore hypothesized that, in normal weight human subjects, a short-term sucrose overfeeding associated with a high-protein, low-fat intake would blunt intrahepatocellular and intramyocellular lipid storage compared to the same sucrose overfeeding associated with a low-protein, high-fat diet. To assess this hypothesis, we carried out a randomized, cross-over controlled trial in 12 healthy male and female subjects. We monitored IHCL and IMCL, postprandial energy expenditure (EE), and blood metabolite concentrations at baseline, i.e. after 3 days on a 10% sucrose weight maintenance diet (WM), and after 6-days overfeeding with 50% extra-energy added as 40% sucrose and 10% lactose with either a high protein-low fat (HP-LF) or a low protein-high fat (LP-HF) content. 

## 2. Materials and Methods

### 2.1. Subjects

Twelve healthy and non-obese volunteers (6 males, mean age 21 ± 1 years, weight 71.6 ± 2.3 kg, BMI 22.5 ± 0.8 kg/m^2^; 6 females mean age 23 ± 1 years, weight 57.3 ± 0.8 kg, BMI 21.2 ± 0.7 kg/m^2^) were included in this study. Volunteers were recruited through advertisements posted at the University of Lausanne and the Lausanne University Hospital. All volunteers were sedentary (less than 2 h of strenuous physical activity per week), were nonsmokers, had no lactose intolerance as documented by a lactose hydrogen breath test [18], and did not take any medication, (except for contraceptive agents which were used by all female participants). They all provided informed written consent. 

### 2.2. Experimental Protocol

The experimental protocol was approved by the ethical committee (Commission d’éthique pour la recherche humaine de l’Etat de Vaud, Switzerland), and was registered at clinicaltrials.gov (NCT02168218). All procedures were performed in accordance with the 1983 revision of the Declaration of Helsinki. The primary outcome of the study was whole body protein turnover using labelled leucine, and will be reported separately. IHCL, IMCL and EE, which are the main focus of this paper, were all secondary outcomes. The experimental protocol is presented in Figure 1.

### 2.3. Dietary Interventions

All participants were studied on two occasions, each one consisting of a 3-day (D-3–D-1) weight-maintenance (WM), low sucrose diet followed by 6-day of sucrose + lactose overfeeding (D1–D6). On one occasion this overfeeding consisted of a 5% dietary protein and 25% fat content; on the other occasion, it was comprised of 20% dietary protein and 10% fat content. The dietary conditions were applied according to a randomized, cross-over design (Figure 1). Randomization was performed according to a pre-defined sequence, which was generated using R, version 3.0.1. (R Foundation for Statistical Computing, Vienna, Austria). The intervention was not blinded due to the nature of the drinks consumed. The two interventions were separated by a washout period of four to eight weeks. 

WM diets were prepared from market foods and provided 100% of energy requirements (estimated from basal energy expenditure, calculated with the Harris-Benedict equation, times a physical activity level of 1.5). Food intake was partitioned into 3 meals/day and 2 snacks/day. It contained 45% of total energy as starch, 10% as sucrose, 33% as lipid, and 12% as protein and 22.6 ± 0.9 g dietary fiber/day; beverages were provided *ad libitum* as water. Overfeeding was attained by adding an extra 50% energy to the weight-maintenance energy requirements, in the form of six drinks per day. Drinks were prepared with skimmed milk and sucrose for the HP-LF condition or with water, lactose, and sucrose for the LP-HF condition, and had a volume of 218 ± 52 ml each. Solid diets were adjusted to obtain the same total energy (150% energy requirement): starch (29%), sucrose (34%) and lactose (7%) in both diets, with 20% protein (2.7 g/kg/day) and 10% fat in HP-LF or 5% protein (0.8 g/kg/day) and 25% fat in LP-HF. The addition of fat in LP-HF was mainly achieved by the addition of olive oil, butter, sauces, and cereals bars. Water consumption was left *ad libitum*. The detailed compositions of all three diets are shown in Table 1.

During each intervention, participants came to the metabolic unit of the Physiology Department of the University of Lausanne to consume their breakfasts, lunches, dinners, and three supplemental drinks under supervision. Every day, they also received two packages of snacks, together with three supplemental drinks during the overfeeding periods to consume between main meals, and were instructed not to consume any other food or drinks except plain water. 

### 2.4. Measurements of IHCL and IMCL

For each intervention, IHCL and IMCL were measured at 4:00 pm on the 3rd day (D-1) on the WM diet (WM_LP-HF_ and WM_HP-LF_) and on the 6th day (D6) on the hypercaloric diets (HP-LF and LP-HF). IHCL and IMCL content were determined by ^1^H-MRS using a clinical 3T MR system (Verio, Siemens Medical, Germany) using methods similar to those described previously for IMCL [19,20] and for IHCL [21]. For the latter, quantification was based on the unsuppressed water signal corrected for transverse relaxation (characterized by the T_2_ value) as determined in each subject individually. Since T_2_ values were found to be significantly different before (WM_LP-HF_, WM_HP-LF_) versus after the diets (LP-HF, HP-LF), but did not differ between diets (LP-HF vs. HP-LF), individually averaged T_2_ values for pre- and post-diet sessions were used for IHCL quantification. Results were expressed as mmol/kg ww.

### 2.5. Metabolic Tests

On days following IHCL and IMCL measurements (D0 and D7), participants were asked to arrive in the fasting state at the Metabolism, Nutrition and Physical Activity Research Center of the Department of Physiology of the University of Lausanne at 7:00 am for a metabolic test (schema shown in Figure 2). They had performed a 24-h urine collection the day before. 

This metabolic test aimed at comparing their fasting and postprandial energy expenditure, plasma hormones, and substrate profiles during periods of weight maintenance and overfeeding. At their arrival, participants were asked to void and discard their urine. They were then weighed and transferred to a bed where they remained in a semi-recumbent position for the next 7.5 h. A catheter was inserted into an antecubital vein for blood collection. Subjects remained fasted for the initial 2.5 h. Four fasting blood samples and a urine collection were obtained during this period. Thereafter, they received two meals, one at 150 min and the second one at 330 min. Meal composition corresponded to the current intervention (i.e., WM on D0 and either HP-LF or LP-HF on D7). The sum of these two meals contained 40% (30% in first and 10% in the second meal) of total daily energy intake, which corresponded to 40% of energy requirements with WM, and to 60% of daily energy requirements during overfeeding periods (HP-LF and LP-HF). Postprandial blood samples were collected at the times 210 min, 270 min, 330 min, 390 min, and 450 min. Respiratory gas exchanges were monitored throughout the experiment by open-circuit indirect calorimetry (Quark RMR, version 9.1b, Cosmed, Rome, Italy), except for brief interruptions during meals. A second urine collection was obtained at the end of the test (time 450 min). Energy expenditure (EE) was calculated using the equations of Livesey and Elia [22].

### 2.6. Analytical Procedures

Plasma glucose, triglycerides (TG), lactate, and urine urea were measured by enzymatic methods (Randox Laboratories, Crumlin, County Antrim, UK). Plasma fructose concentrations were measured by GC–MS apparatus (Agilent Technologies, Santa Clara, CA, USA) [23]. Insulin and glucagon were assessed by radioimmunoassays (Millipore, Billerica, MA, USA). Plasma lipoprotein subfractions were separated by ultracentrifugation [24].

### 2.7. Statistical Analysis

All results are expressed as means ± SEMs. Postprandial results for all parameters (except for IGF1 and glucagon, which were determined in fasting conditions at only 2-time points postprandial) were expressed as the incremental area under the curve (iAUC _(0-300 min)_), which was obtained using the trapezoidal method by subtracting the fasting value. As a preliminary analysis, the normality of data was checked with Shapiro-Wilk tests for all parameters analyzed. Non-normally distributed data were log-transformed (IHCL, fasting insulin, glucagon, TG, and postprandial glucagon). Two-way ANOVA assessed the effects of overfeeding, protein/fat content (HP-LF vs. LP-HF), and interaction between overfeeding x protein/fat content with repeated measures. Tukey post hoc tests were performed to compare individuals when needed. All statistical analyses were performed using Prism 7 (GraphPad Software, Inc., La Jolla, USA). The number of subjects included in the study was based on a power analysis related to whole body protein turnover (not reported here).

## 3. Results

The recruitment and follow up of subjects took place between June 2013 and April 2016. All volunteers completed the investigation and reported that they did not take any additional caloric drinks and food during the study. One volunteer was not included in the calculation of postprandial fructose due to missing plasma samples. Two volunteers were excluded from 24 h urinary concentration, excretion, and clearance calculation due to missing urine collections. All other calculations were performed with all 12 volunteers.

### 3.1. Fasting Condition 

Fasting parameters are shown in Table 2. All fasting parameters were not significantly different after WM_LP-HF_ and WM_HP-LF_. Body weight increased by 0.7 ± 0.1 kg (males 0.9 ± 0.2 kg, females 0.6 ± 0.1 kg) between D0 and D7 after LP-HF and by 1.4 ± 0.2 kg after HP-LF (males 1.8 ± 0.2 kg, females 0.9 ± 0.1 kg) (for the whole group: *p* < 0.001 for overfeeding, *p* > 0.999 for protein/fat content, *p* = 0.009 for overfeeding × protein/fat content). Fasting EE increased from 1.11 ± 0.06 kcal/min (WM_LP-HF_) to 1.12 ± 0.05 kcal/min (LP-HF), and from 1.10 ± 0.05 kcal/min (WM_HP-LF_) to 1.18 ± 0.05 kcal/min (HP-LF), (*p* = 0.018 for overfeeding, *p* = 0.126 for protein/fat content, *p* = 0.024 overfeeding x protein/fat content). Fasting plasma glucose, fructose, lactate, TG, and insulin all increased to the same extent with HP-LF and LP-HF (Table 2). Fasting plasma NEFA decreased to the same extent with HP-LF and LP-HF. In contrast, fasting glucagon concentration and IGF-1 concentrations increased with HP-LF, but remained stable (glucagon) or slightly decreased (IGF-1) with LP-HF.

### 3.2. IHCL and IMCL Concentrations

IHCL and IMCL concentrations after WM and after LP-HF and HP-LF are shown in Figure 3. No statistically significant difference was observed between WM_LP-HF_ and WM_HP-LF_. Compared to WM conditions, IHCL and IMCL concentrations increased significantly with both LP-HF and HP-LF overfeeding. However, IHCL increased more importantly with LP-HF than with HP-LF (*p* < 0.001 for effect of overfeeding, *p* < 0.001 for effect of dietary protein/fat content, and *p* < 0.001 for interaction overfeeding x protein/fat content). IMCL also increased more with LP-HF than with HP-LF (*p* < 0.001 for overfeeding, *p* = 0.025 for protein/fat content, and *p* = 0.002 for overfeeding x protein/fat content). 

### 3.3. Postprandial Parameters

Postprandial metabolic parameters were not significantly different after WM diets. Postprandial EE and diet-induced thermogenesis were both significantly higher with LP-HF and HP-LF than under their respective WM conditions. Furthermore, EE increased more after HP-LF (from 1.23 ± 0.05 to 1.55 ± 0.06 kcal/min) than after LP-HF (from 1.24 ± 0.05 to 1.41 ± 0.06 kcal/min) (*p* < 0.001 for overfeeding, *p* = 0.013 for protein/fat content, and *p* < 0.001 for overfeeding x protein/fat content).

The postprandial iAUCs for blood metabolites and hormones are shown in Table 3. Postprandial blood glucose did not significantly change with HP-LF and LP-HF compared to their respectively WM conditions. Postprandial fructose, lactate, TG, and insulin iAUC were significantly higher in HP-LF and LP-HF than in the respective WM conditions.

HP-LF and LP-HF nonetheless differentially altered postprandial insulin, fructose, and lactate concentrations: HP-LF increased postprandial insulin concentrations more than LP-HF, but decreased postprandial fructose and lactate (see Table 3 for detailed statistics). Postprandial plasma uric acid concentration, measured at time 450 min, decreased from 0.38 ± 0.02 (WM) to 0.30 ± 0.02 mmol/L with HP-LF, but increased from 0.38 ± 0.03 (WM) to 0.42 ± 0.04 mmol/L with LP-HF (*p* = 0.283 for overfeeding, *p* = 0.001 for diet, *p* < 0.001 for overfeeding × protein/fat content). Plasma glucagon, measured at time 450 min, increased from 62.6 ± 3.3 to 87.9 ± 8.4 pg/mL with HP-LF, but did not change with LP-HF: 65.1 ± 4.4 vs. LP-HF: 70.6 ± 4.8 pg/mL, (*p* < 0.001 for overfeeding, *p* = 0.026 for protein/fat content, and *p* = 0.001 for overfeeding x protein/fat content). 

24-h urinary excretion and clearance of creatinine and uric acid are shown in Table 4. LP-HF and HP-LF did not significantly change 24-h urinary excretion and clearance of creatinine. HP-LF increased urinary excretion of uric acid and uric acid clearance while LP-HF decreased it. Compared to LP-HF, HP-LF significantly increased urinary creatinine and uric acid clearance; it also increased total 24-h uric acid excretion.

## 4. Discussion

This study was designed to assess whether the consequences of sucrose overfeeding differ according to concomitant changes in daily protein and fat intake. Our main findings were that both HP-LF and LP-HF increased IHCL, IMCL, and blood triglycerides concentrations, but increments were reduced on average by 78% for IHCL and by 59% for IMCL with HP-LF compared to LP-HF. In addition, fasting and postprandial EE were significantly higher with HP-LF than LP-HF. However, blood triglyceride concentrations were not significantly different with HP-LF and LP-HF. Finally, blood uric acid concentrations were increased with LP-HF, but decreased with HP-LF.

Our experimental design compared the effects of two hypercaloric high sucrose diets, one with a high protein-low fat content and the other with a low protein-high fat content, to that of a weight maintenance control diet. All three diets contained an amount of starch equivalent to approximately 45% total energy requirements, and the two hypercaloric diets contained 150% of daily energy requirements, with about 50% of energy requirements as sucrose, and 7% of energy requirements as lactose. Lactose intake was higher in HP-LF than in WM because of a high milk protein intake and was balanced by lactose addition in LP-HF in order to have equal carbohydrate amounts and composition in both diets. Dietary saturated-monounsaturated and polyunsaturated fatty acid proportions were also different in each diet.

The dietary composition had a profound effect on the amount of ectopic lipids being deposited during overfeeding. HP-LF and LP-HF both increased lipid storage in the liver and muscle, two sites in which ectopic lipid deposition is known to be associated with adverse long-term effects [1]. Several short-term studies had previously documented that excess energy intake from fructose or glucose increased IHCL [10,25,26] and IMCL [26,27,28]. In our study, this effect was most notable in the liver, where IHCL increased by 542 ± 105% after LP-HF. It was milder in skeletal muscle, where we nonetheless observed a significant increase of +24 ± 3% after LP-HF. In both sites, the increases induced by HP-LF were significantly lower than those induced by LP-HF. Excess energy intake from sugars is thought to increase IHCL by enhancing hepatic *de novo* lipogenesis and inhibiting intrahepatic lipid oxidation [29]. Several hypotheses can be proposed to account for the differential effects of HP-LF and LP-HF. First, LP-HF contained more lipids than HP-LF. Previous experiments have shown that fat overfeeding increases IHCL synthesis from intestinally derived TG-rich lipoprotein particles and/or circulating NEFA [13,30,31]. It has also been shown that fructose and fat have additive effects on IHCL during combined fructose-fat overfeeding [10]. It is therefore likely that, with LP-HF, the high dietary sugar and fat intake had additive effects on IHCL. Second, dietary protein may decrease IHCL independently of dietary fat or energy intake. In support of this hypothesis, a former study reported that IHCL were increased in healthy subjects fed a hypercaloric, high fat diet containing 130% energy requirements. However, the addition of protein to this high fat diet resulted in a similar daily fat and carbohydrate intake, but also in a higher total energy and protein intake with significantly reduced IHCL [13]. The mechanisms by which an increased protein intake may reduce IHCL remain unknown. Inhibition of *de novo* lipogenesis has been postulated [13], but fractional hepatic *de novo* lipogenesis was stimulated to the same extent in healthy subjects overfed with fructose alone or with fructose and proteins [16]. A stimulation of hepatic VLDL-TG secretion and extrahepatic VLDL-TG clearance [16], or a protein-induced increase in plasma bile acid concentrations [13] have also been proposed to play a role. In contrast, no effect of dietary protein intake on IMCL has been reported to our knowledge. Finally, changes in dietary fatty acids composition may modulate diet-induced hepatic fat deposition (reviewed in reference [32]). Hepatic steatosis in animal models is readily produced by consumption of a high saturated fat diet with low PUFA content. In contrast, there is evidence that PUFA or oleic acid supplementation may actually blunt diet-induced hepatic steatosis [32]. In the present study, dietary protein intake in HP-LF was increased through the consumption of skimmed dairy products to avoid an increase in SFA, and dietary fat intake in LP-HF was increased by consumption of vegetable oils (mainly olive oil). As a result, total daily SFA intake was only slightly higher in LP-HF than in HP-LF (34.7 ± 1.5 vs. 20.4 ± 0.9 g/day) while MUFA+PUFA intake was markedly increased. It is therefore unlikely that the higher IHCL observed with LP-HF can be explained by the differences in dietary fat composition.

The postprandial increases in plasma TG concentrations were 5-fold higher with HP-LF and 4-fold higher with LP-HF than with WM. Several studies have reported that fructose and sucrose overfeeding increases fasting and postprandial blood triglyceride by increasing hepatic *de novo* lipogenesis and VLDL-TG secretion and by decreasing the postprandial clearance of triglyceride-rich lipoprotein particles [27,33,34]. It is therefore likely that an upregulation of lipogenic enzymes with sucrose overfeeding contributed to this hypertriglyceridemia. However, the meals administered during the metabolic tests contained 50% more total energy in overfeeding than in weight-maintenance control conditions, and, therefore, contained also more sucrose and fat, which makes it difficult to sort out the relative role of sucrose and other macronutrients. Globally, the increase in postprandial TG concentrations was not significantly different in HP-LF and LP-HF. 

The effect of overfeeding on energy expenditure was also markedly dependent on dietary composition. Postprandial EE increased significantly with both HP-LF and LP-HF, mainly due to the fact that the test meals ingested in both conditions had a caloric content 50% higher than in the control weight-maintenance condition. Postprandial EE increased more with HP-LF than LP-HF. This is most likely explained by the high energy cost of amino-acid metabolism [35]. 

We also assessed whether dietary composition had significant effects on postprandial blood metabolic markers during overfeeding. The total carbohydrate and sucrose content of meals ingested during the metabolic tests were higher in overfeeding than in the WM control condition, and postprandial increments in blood fructose, lactate, and insulin were accordingly enhanced. Similarly, postprandial NEFA was decreased to lower levels in overfeeding than in WM conditions. However, postprandial blood glucose responses were not significantly altered. Most postprandial parameters were not significantly different in HP-LF and LP-HF overfeeding. However, postprandial glucagon increased more with HP-LF than with LP-HF, as expected due to the well-known stimulation of glucagon secretion by circulating amino-acids after protein ingestion [36]. Surprisingly, blood fructose and lactate concentration increased less with HP-LF than LP-HF. It is possible that the lower lactate concentration was secondary to glucagon stimulating hepatic lactate uptake [37]. The lower fructose response was unexpected, however, and may suggest that hepatic fructose extraction was enhanced when consumed with proteins. Nutrient- or glucagon-mediated changes in portal blood flow may also be implicated [38]. Alternatively, it is possible that gastric emptying was delayed with HP-LF meals, thus accounting for a slower fructose absorption [39]. Finally, compared to WM, postprandial increases in uric acid were higher with LP-HF, but lower with HP-LF, while urinary uric acid excretion and uric acid clearance were significantly increased with HP-LF. This suggests that both HP-LF and LP-HF increased uric acid production, possibly due to the fructose component of sucrose [40], and that an increase in glomerular filtration rate, possibly mediated by glucagon [41], increased uric acid excretion, thus preventing an increase in blood uric acid. Elevated lactate concentrations are also known to impair renal uric acid clearance [42], and it is, therefore, possible that lower lactate concentrations during HP-LF than LP-HF overfeeding also played a role. Our data, however, do not allow accurate comparisons of uric acid production and excretion between HP-LF and LP-HF.

The present study limitations need to be acknowledged. First, we did not include isotopic measurements of *de novo* lipogenesis and VLDL-TG kinetics, and therefore cannot identify the mechanisms by which HP-LF decreased IHCL and IMCL compared to LP-HF. Second, not only total dietary fat intake, but also the proportions of SFA-MUFA-PUFA were different between diets, and we cannot exclude the possibility that this may have impacted IHCL or IMCL storage. Third, in HP-LF condition, dietary protein content was increased by addition of dairy products; whether the observed effects are generic to dietary proteins or specific to dairy products remains to be evaluated. Finally, our study was of short duration and was limited to a small group of healthy male and female subjects, and results may not apply to other subgroups of the population (e.g., overweight subjects or subjects with the metabolic syndrome). 

## 5. Conclusions

In summary, our data indicate that overfeeding with a high sucrose, high protein/low-fat diet markedly reduces ectopic fat accumulation in the liver and muscle, and increases energy expenditure, compared to an isocaloric overfeeding with high sucrose, low protein/high-fat diet. This may be due to an additive effect of sucrose and dietary fat and/or a protective effect of dietary protein on ectopic fat accumulation. 

## Figures and Tables

**Figure 1 nutrients-11-00209-f001:**
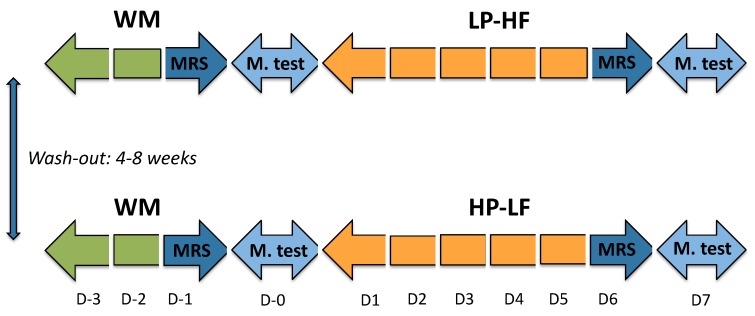
Experimental protocol. Each participant took part in two overfeeding periods according to a randomized, cross-over design. WM: weight maintenance diet, LP-HF: hypercaloric (150% energy requirement high-sucrose, low protein-high fat); HP-LF: hypercaloric (150% energy requirement high-sucrose, high protein-low fat); MRS: magnetic resonance spectroscopy for measurement of IHCL and IMCL; M. test: metabolic test, consisting of measurements of energy expenditure, plasma hormones, and substrate concentrations after ingestion of WM meal providing 40% of total energy requirements (D0), or LP-HF/HP-LF meals providing 60% of total energy requirements.

**Figure 2 nutrients-11-00209-f002:**
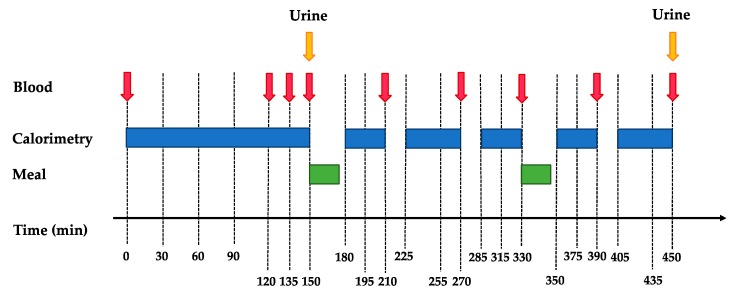
Schema of metabolic tests at D0 and D7.

**Figure 3 nutrients-11-00209-f003:**
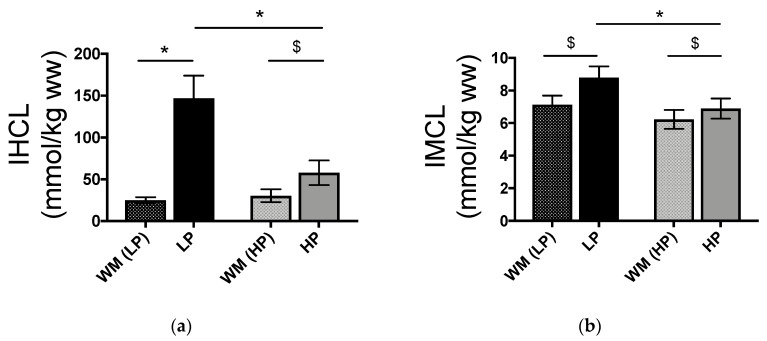
Intrahepatocellular (IHCL) (**a**) and intramyocellular (IMCL) lipids (**b**) in response to weight maintaining diet (WM_LP-HF_ and WM_HP-LF_) and overfeeding with LP-HF and HP-LF. *n* = 12; significant responses from WM_LP-HF_ and WM_HP-LF_ were measured by 2-way ANOVA for repeated measures with interaction. *: *p* < 0.001, interaction overfeeding × protein/fat content. $: *p* < 0.005, Tukey post hoc tests.

**Table 1 nutrients-11-00209-t001:** Energy content and macronutrient composition of WM, LP-HF and HP-LF.

Diet Composition	WM		LP-HF			HP-LF		
	Solid Diet kcal/day (%)	Beverages kcal/day (%)	Solid Diet kcal/day (%)	Beverages kcal/day (%)	Total LP-HF kcal/day (%)	Solid Diet kcal/day (%)	Beverages kcal/day (%)	Total HP-LF kcal/day (%)
Starch	1061 (45)	-	1054	-	1054 (29)	1043	-	1043 (29)
Sucrose	249 (10)	-	241	965	1206 (34)	246	964	1210 (34)
Lactose	-	-	-	245	245 (7)	-	246	246 (7)
Protein	274 (12)	-	194	-	194 (5)	514	178	692 (20)
Fat	781 (33)	-	886	-	886 (25)	357	12	369 (10)
SFA	263 (34)	-	313	-	313 (35)	184	-	184 (52)
MUFA	280 (36)	-	389	-	389 (44)	102	-	102 (29)
PUFA	202 (26)	-	168	-	168 (19)	55	-	55 (15)
**Total kcal**	2365	-	2375	1210	3585	2160	1400	3560

WM: weight maintenance diet; LP-HF: high-sucrose, low-protein; HP-LF: high-sucrose, high-protein. Data are expressed as kcal/day; values into bracket represent % of total energy intake. For SFA, MUFA and PUFA, values in () are given as % total fat intake.

**Table 2 nutrients-11-00209-t002:** Fasting plasma metabolites and hormones concentrations.

Fasting	WM (LP-HF)	LP-HF	WM (HP-LF)	HP-LF	*p* Value		
					Overfeeding	Protein/Fat Content	OxP
Glucose (mmol/L)	4.56 ± 0.07	4.78 ± 0.07	4.46 ± 0.11	4.76 ± 0.09	<0.001	0.444	0.383
Fructose (μmol/L)	25.95 ± 1.41	27.15 ± 1.49	26.35 ± 1.37	28.0 ± 1.32	0.022	0.611	0.750
Lactate (mmol/L)	0.70 ± 0.06	1.22 ± 0.07	0.64 ± 0.04	1.16 ± 0.09	<0.001	0.107	0.935
Uric acid (mmol/L)	0.38 ± 0.02	0.38 ± 0.03	0.39 ± 0.02	0.30 ± 0.02	<0.001	<0.001	<0.001
TG (mmol/L)	0.68 ± 0.07	1.54 ± 0.22	0.66 ± 0.08	1.68 ± 0.19	<0.001	0.429	0.119
NEFA (mmol/L)	0.72 ± 0.05	0.44 ± 0.09	0.77 ± 0.04	0.37 ± 0.07	<0.001	0.779	0.097
Insulin (μU/mL)	8.42 ± 0.83	10.95 ± 1.02	7.82 ± 0.79	11.73 ± 1.54	<0.001	0.744	0.295
Glucagon (pg/mL)	72.42 ± 4.83	72.49 ± 4.94	68.67 ± 4.03	79.27 ± 5.35	0.059	0.276	0.036
IGF-1 (ng/mL)	212 ± 13	176 ± 12	174 ± 18	208 ± 13	0.901	0.712	<0.001

WM: weight maintenance diet; LP-HF: high-sucrose, low-protein; HP-LF: high-sucrose, high-protein. All values are mean ± SEM, *n* = 12. A significant difference in each condition, *p* < 0.05 (2-way ANOVA with repeated measures). OxP: Overfeeding x protein/fat content.

**Table 3 nutrients-11-00209-t003:** Metabolites and hormones at postprandial states.

Postprandial	WM (LP-HF)	LP-HF	WM (HP-LF)	HP-LF	*p* Value		
					Overfeeding	Protein/Fat Content	OxP
iAUC Glucose (mmol/L*300min)	504.0 ± 40.5	495.3 ± 69.2	560.3 ± 43.7	471.4 ± 57.2	0.242	0.616	0.189
iAUC Fructose (mmol/L*300min)	4.2 ± 0.3	30.3 ± 2.9	4.8 ± 0.5	23.4 ± 2.2	<0.001	0.005	0.003
iAUC Lactate (mmol/L*300min)	78.8 ± 12.8	239.8 ± 24.5	92.2 ± 15.1	139.3 ± 15.7	<0.001	0.001	0.001
iAUC TG (mmol/L*300min)	29.3 ± 6.8	121.3 ± 15.3	24.2 ± 8.0	126.7 ± 16.9	<0.001	0.986	0.471
iAUC NEFA (mmol/L*300min)	−162.6 ± 12.5	−86.7 ± 23.2	−173.1 ± 11.0	−76.6 ± 17.2	<0.001	0.984	0.051
iAUC Insulin (μU/mL*300min)	11378 ± 1232	19228 ± 1708	11138 ± 1488	24123 ± 2790	<0.001	0.061	0.028

WM: weight maintenance diet; LP-HF: high-sucrose, low-protein; HP-LF: high-sucrose, high-protein. All values are mean ± SEM, *n* = 12. A significant difference in each condition, *p* < 0.05 (2-way ANOVA, with repeated measures). OxP: Overfeeding x protein/fat content. In the calculation of iAUC fructose (*n* = 11) one volunteer was excluded for reason of missing plasma data.

**Table 4 nutrients-11-00209-t004:** 24-h urinary creatinine and uric acid excretion and clearance.

	WM (LP-HF)	WM (HP-LF)	LP-HF	HP-LF	*p* Value		
					Overfeeding	Protein/Fat Content	OxP
24-h urinary excretion							
Creatinine (mmol/24h)	13.6 ± 1.8	12.6 ± 1.2	13.0 ± 0.8	12.7 ± 1.1	0.264	0.450	0.638
Uric acid (mmol/24h)	3.5 ± 0.2	3.3 ± 0.2	3.3 ± 0.2	4.1 ± 0.4	0.049	0.238	0.022
Urinary clearance rate							
Creatinine (ml/min)	129.8 ± 9.4	133.7 ± 9.7	135.1 ± 10.8	153.0 ± 13.0	0.279	0.309	0.282
Uric acid (ml/min)	6.9 ± 0.6	6.5 ± 0.6	6.1 ± 0.4	10.0 ± 1.4	0.005	0.015	0.004

WM: weight maintenance diet; LP-HF: high-sucrose, low-protein; HP-LF: high-sucrose, high-protein. All values are mean ± SEM, *n* = 10 as two volunteers were excluded because of missing samples. A significant difference in each condition, *p* < 0.05 (2-way ANOVA, with repeated measures). OxP: Overfeeding x protein/fat content.

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
