# Peer review of "Effects of Dietary Protein and Fat Content on Intrahepatocellular and Intramyocellular Lipids during a 6-Day Hypercaloric, High Sucrose Diet: A Randomized Controlled Trial in Normal Weight Healthy Subjects"

_nutrients, 2019, doi:10.3390/nu11010209_

Round 1

Reviewer 1 Report

In this study, authors studied the effects of a high-sucrose hypercaloric diets that are different in protein content (5% energy vs. 20% energy) and fat content (25% energy vs. 10% energy) on intrahepatocellular and intramyocellular lipids. Authors concluded that a high dietary protein intake reduces sucrose-induced liver and muscle fat deposition. Authors have conducted a study that is not easy to carry on. However, the conclusion is not supported by the results because it is hard to differentiate whether the differences observed are due to differences in protein content or fat content. Authors need to revise the conclusion and discuss the effects of difference in fat content as well rather than focusing on the effects of protein content difference.

 Major issues

There are differences in both protein and fat contents between LP and HP diets. However, authors presented the results and discussed as if the effects observed were mainly due to the difference in protein content. Discussion needs to include the possible effects of fat content difference because dietary fat content can have significant impact on the lipid profile.

Anthropometric data of the patients should be presented (male and female separately). Weight change results are needed.

Regarding macronutrient composition of the diets, fatty acid compositions are needed.

Conclusion should be changed. Conclusions in the abstract and the text are not supported by the results and seems like an overstatement.

Author Response

 Major issues

1.      There are differences in both protein and fat contents between LP and HP diets. However, authors presented the results and discussed as if the effects observed were mainly due to the difference in protein content. Discussion needs to include the possible effects of fat content difference because dietary fat content can have significant impact on the lipid profile.

Answer: We agree that the observed effects cannot be unequivocally attributed to dietary protein content, and that both a high fat intake and a low protein content may have enhanced the effects of fructose overfeeding. We have now accordingly changed the title of the article to “Effects of dietary protein and fat content on intrahepatocellular and intramyocellular lipids during in a 6-day hypercaloric, high sucrose diet: a randomized controlled trial in normal weight healthy subjects” , and the conclusions to “In summary, our data indicate that overfeeding with a high sucrose, high protein/low-fat diet markedly reduces ectopic fat accumulation in the liver and muscle, and increases energy expenditure, compared to an isocaloric overfeeding with high sucrose, low protein/high-fat diet. This may be due to an additive effect of sucrose and dietary fat and/or a protective effect of dietary protein on ectopic fat accumulation”

2.      Anthropometric data of the patients should be presented (male and female separately). Weight change results are needed.

Answer: Anthropometric data and weight changes have now been given for male and female separately

3.      Regarding macronutrient composition of the diets, fatty acid compositions are needed.

Answer: Dietary fatty acid composition has now been detailed in table 1. Its possible impact on results has been briefly discussed.

4.      Conclusion should be changed. Conclusions in the abstract and the text are not supported by the results and seems like an overstatement.

Answer: Conclusions have been changed

Reviewer 2 Report

Congratulations to the authors for a very serious piece of work.  In that light my comments are for the purpose of increasing the impact of the paper.

.  I think the paper could benefit from editing.  The English could be improved, but more importantly too much data is included.  The main point is figure 2, and everything else is extra, but there is so much more  data presented the  figure is somewhat lost.  For example, the large Also, the DIT data are predictable.  table with urea data is not needed in my opinion.  With regard to fig 2, it is not clear if the high protein diet still has a significant increase in liver fat- it doesn't look like it, but if I understand the notation correctly you are showing a significant difference. 

Importantly, I the conclusion you say that the HP diet may result in lower body weight over time.  The difference in energy expenditure between diets is not large enough to make that conclusion.   

Author Response

1.      I think the paper could benefit from editing.  The English could be improved, but more importantly too much data is included. 

Answer: Done

2.      The main point is figure 2, and everything else is extra, but there is so much more  data presented the  figure is somewhat lost.  For example, the large Also, the DIT data are predictable.  table with urea data is not needed in my opinion.

Answer: We have now deleted the EE-thermogenesis fig (old Fig3) from the results, and included effects of EE directly in the text. We have also simplified urinary data presentation by showing only uric acid and creatinine excretion and clearance (relevant to different effects of diets on blood uric acid) in table 4 . Calculation of urinary clearances has also been deleted from the methods since it is standard procedure

3.      With regard to fig 2, it is not clear if the high protein diet still has a significant increase in liver fat- it doesn't look like it, but if I understand the notation correctly you are showing a significant difference. 

Answer: Yes, both high protein and low protein diets significantly increase IHCL, has indicated in fig 2. The effect (IHCL increment) is however significantly larger with HP than LP.

4.      Importantly, I the conclusion you say that the HP diet may result in lower body weight over time.  The difference in energy expenditure between diets is not large enough to make that conclusion.  

Answer: We agree that this conclusion was not sufficiently supported by our present data, and have now deleted it.

Round 2

Reviewer 1 Report

Issues raised have been addressed.

Author Response

N/A